# Multifaceted Regulation of PTEN Subcellular Distributions and Biological Functions

**DOI:** 10.3390/cancers11091247

**Published:** 2019-08-26

**Authors:** Tian Liu, Yiwei Wang, Yubing Wang, Andrew M. Chan

**Affiliations:** 1School of Biomedical Sciences, Room 705, Lo Kwee-Seong Integrated Biomedical Sciences Building, The Chinese University of Hong Kong, Hong Kong SAR, China; 2School of Bioscience and Technology, Weifang Medical University, Weifang 261053, China

**Keywords:** PTEN, PI3K, phosphatase, regulation

## Abstract

Phosphatase and tensin homolog deleted on chromosome 10 (*PTEN*) is a tumor suppressor gene frequently found to be inactivated in over 30% of human cancers. *PTEN* encodes a 54-kDa lipid phosphatase that serves as a gatekeeper of the phosphoinositide 3-kinase pathway involved in the promotion of multiple pro-tumorigenic phenotypes. Although the PTEN protein plays a pivotal role in carcinogenesis, cumulative evidence has implicated it as a key signaling molecule in several other diseases as well, such as diabetes, Alzheimer’s disease, and autism spectrum disorders. This finding suggests that diverse cell types, especially differentiated cells, express PTEN. At the cellular level, PTEN is widely distributed in all subcellular compartments and organelles. Surprisingly, the cytoplasmic compartment, not the plasma membrane, is the predominant subcellular location of PTEN. More recently, the finding of a secreted ‘long’ isoform of PTEN and the presence of PTEN in the cell nucleus further revealed unexpected biological functions of this multifaceted molecule. At the regulatory level, PTEN activity, stability, and subcellular distribution are modulated by a fascinating array of post-translational modification events, including phosphorylation, ubiquitination, and sumoylation. Dysregulation of these regulatory mechanisms has been observed in various human diseases. In this review, we provide an up-to-date overview of the knowledge gained in the last decade on how different functional domains of PTEN regulate its biological functions, with special emphasis on its subcellular distribution. This review also highlights the findings of published studies that have reported how mutational alterations in specific PTEN domains can lead to pathogenesis in humans.

## 1. Introduction 

Phosphatase and *TEN*sin homolog deleted on chromosome 10 (*PTEN*) was originally discovered as a bona fide tumor suppressor gene on human chromosome 10q23.3, a region frequently lost in prostate cancer and glioblastoma multiforme [1,2,3]. In the subsequent decades, intensive surveys on *PTEN* mutations in human cancers have revealed widespread genetic and epigenetic inactivation of this gene. The rate of point mutations varies between different tumor types, being as high as 37% in human endometrial cancer [4,5]. Loss of heterozygosity is frequently responsible for the complete inactivation of a tumor suppressor gene (*NF1* or *TP53*). *PTEN* is unique in the respect that the loss of its single allele can lead to carcinogenesis. This haploinsufficiency of the *PTEN* gene has been shown to promote prostate cancer [6,7]. Furthermore, the *PTEN* gene expression is subject to modulation by a host of noncoding RNAs (ncRNAs) in various human cancers [8]. These ncRNAs include more than a dozen microRNAs (miRNAs) and long ncRNAs. Dysregulated interactions between miRNAs and competing endogenous RNAs that share similar miRNA response elements can lead to reduced PTEN expression and promote tumor progression [9].

Inactivation of PTEN’s biochemical function appears to be the major mechanism underlying cancer pathogenesis. PTEN is a dual-specificity lipid and protein phosphatase. It mediates the dephosphorylation of the 3′ phosphate of phosphatidylinositol 3,4,5-triphosphate (PIP3) to phosphatidylinositol 4,5-bisphosphate (PIP2) [10] and dephosphorylates itself at threonine 366 (T366), thereby unmasking its ability to inhibit cell invasion [11]. Numerous signaling molecules have been shown to physically interact with PTEN (Figure 1). Proteins that are known to be the direct substrates of PTEN include PTK6, UBB, AKT1, PLK1, RAB7, IKBKB, IRS1, and CREB1 [12]. Overall, PTEN plays a pivotal role as a gatekeeper of the phosphoinositide 3-kinase (PI3K) pathway and represses downstream signaling events that control cell proliferation, cell survival, and protein synthesis [13].

The partial crystal structure of PTEN resolved previously has revealed the signature motif HCXXGXXR in its catalytic pocket [14]. The pocket size is larger than that of conventional protein phosphatases, presumably required for accommodating the larger PIP3 substrate. The C-terminal half of PTEN possesses a C2 domain responsible for phospholipid binding and critical for membrane targeting [15]. The tail region of ~90 amino acids is referred to as the PEST domain that is rich in negatively charged amino acid residues [16]. The protein then terminates in a four-amino acid protein–protein interaction domain called the PDZ-binding motif (PDZ-BM) [17,18].

At the evolutionary level, PTEN is a unique protein as it is not closely related to other lipid or protein phosphatases. Orthologs of human *PTEN* gene have been reported in evolutionarily distant organisms, including zebrafish, *Drosophila melanogaster*, *Caenorhabditis elegans*, and *Saccharomyces pombe* (Figure 2). The human PTEN protein has sequence similarity with the human Tensin that encodes a focal adhesion protein linking actin filaments and integrins and the auxilin gene that encodes a C2 domain containing protein tyrosine phosphatase-like molecule [12,14]. Interestingly, all three members of the tensin family—namely, tensin 1, 2, and 3—possess a protein tyrosine phosphatase domain, albeit inactive, in their N-terminus [19]. It is believed that PTEN belongs to a class of molecules containing the protein tyrosine phosphatase (PTP)-C2 superdomain that was formed prior to the fungi, plant, and animal kingdom divergence [19]. This review article will provide an up-to-date overview of the functional domains of PTEN involved in the regulation of its biochemical and biological functions with special emphasis on its role in different subcellular compartments.

## 2. Domains of PTEN

### 2.1. N-Terminal Domain

The 1209-bp coding region of PTEN encodes a 403-amino acid protein with a predicted molecular mass of ~47 kDa. The primary amino acid sequence of PTEN encodes a bipartite molecule and is flanked by flexible protease-sensitive sequences in the amino (N)- and carboxyl (C)-termini. The N-terminal 32-amino acid region is unstructured and possesses three overlapping motifs with distinct biological functions. First, a 10-amino acid PIP2-binding motif (PBM) encompassing residue 6 to 15 that binds either PI(4,5)P2 or PI(5)P to allosterically stimulate the intrinsic lipid phosphatase activity toward PIP3 [10,20]. Three basic residues at K13, K14, and R15 are critical for PTEN activation and are mutated in multiple human tumors (Figure 3) [21]. Second, a nuclear localization sequence of monopartite type between residues 7 and 31, which is characterized by a stretch of basic residues from 13 to 15 (RNKRR). Interestingly, the K13 residue has been shown to undergo both mono- and polyubiquitination [22]. However, the sequence in the vicinity of K13 does not possess the classical consensus sequence Ψ-K-x-E, where Ψ is a large hydrophobic residue, K is the lysine residue being modified, X is any amino acid, and E is glutamate residue [23]. In fact, most online ubiquitination site prediction software failed to register a high score for K13. Nevertheless, K13 has been demonstrated to be ubiquitinated in in vivo assays and has been shown to be responsible for PTEN nuclear import [22]. In addition, a short stretch of sequence from residue 19 to 25 enriched in negatively charged amino acids (DGFDLDL) has been shown to mediate cytoplasmic localization [24].

### 2.2. Catalytic Domain

The N-terminal 179-amino acid region from residue 7 to 185 was originally found to display sequence similarity with the dual-specificity phosphatase (DSP), vaccinia virus VH1-related phosphatase, and the PTP1B [14]. A search in the BLAST protein database revealed transmembrane phosphatase with tensin homology, TPTE, demonstrating the highest level of protein sequence similarities [25]. Similar to other DSPs and PTPs, PTEN harbors a HCXXGXXR signature motif between residues 123 and 130, which is referred to as the P loop located at the bottom region of the active site. Residues C124 and R130 are essential for catalysis, whereas H123 and G127 are critical for the P loop conformation. Mutations are frequently identified in this region [4] (Figure 3). In addition, the D92 residue in the “WPD” loop serves as a general acid to mediate the protonation of the leaving oxygen group. However, there are several structural distinctions. First, the active site pocket in PTEN is ~5 × 11 Å wide, which is two times wider than that of PTPB1, although they have a similar depth of ~8 Å. Second, there is an 11-amino acid insertion between residues 42 and 52 and a 4-amino acid insertion between residues 163 and 166. The latter insertion is referred to as the “TI” loop (because of the conserved threonine and isoleucine residues), and this rigid structure has been suggested to cause the extension of the active site in PTEN [14]. The wider and deeper opening of the active site in PTEN allows accessibility for PIP3, phosphoserine, phosphothreonine, and phosphotyrosine substrates, which reflects the biological versatility of PTEN. Indeed, the G129E mutation found in human tumors highlight the structural determinant of lipid versus protein substrate specificity. G129 is located at the bottom of the active site. The mutation of the glycine residue to glutamate impedes PIP3 access but without affecting protein substrate binding. Thus, G129E mutant is lipid phosphatase dead but protein phosphatase competent [26]. Interestingly, mutagenesis analysis revealed a PTEN Y138L mutant with preserved lipid phosphatase activity but abolished protein phosphatase activity [27]. Thus, PTEN is unique in having dual substrate specificity primarily because of its unique primary coding sequences.

### 2.3. C2 Domain

The C-terminal 166-amino acid region from residue 186 to 351 exhibits structural topology similar to that of the C2 domains of protein kinase Cδ (PKCδ), phospholipase C δ1 (PLCδ1), and phospholipase A2 (cPLA2) [28]. The overall structure comprises two antiparallel β sheets with two short α helices positioned between the two strands [14]. The ability of both PLCδ1 and cPLA2 to bind calcium (Ca^2+^) through their three Ca^2+^-binding loops—namely, Ca^2+^-binding region 1 (CBR1), CBR2, and CBR3—induces a change in the electrical potential that modulates lipid-binding affinity [29]. However, the C2 domain of PTEN lacks all but one of the Ca^2+^-binding motifs and is predicted to not bind Ca^2+^. Instead, the C2 domain of PTEN has a CBR3 loop between residues 259 and 268, and it possesses five positively charged residues at K260, K263, K266, K267, and K269, which interface with the negatively charged groups of phospholipids on the plasma membrane. Furthermore, the CBR3 loop is positioned perpendicularly to the membrane interphase. The presence of two hydrophobic residues at M264 and L265 near the tip of the CBR3 loop is believed to mediate membrane insertion and anchoring of PTEN to the lipid bilayer. Using POPC/POPS anionic vesicles, the C2 domain alone binds with 30 times weaker affinity than the full-length molecule, suggesting that the N-terminal phosphatase domain, not the C2 domain, is critical for driving membrane recruitment [15]. In addition, PTEN does not have a high affinity toward the nuclear membrane [15].

### 2.4. Tail Region

The C-terminal 52-amino acid region between residues 352 and 403 constitutes the tail region of PTEN. It comprises two regulatory motifs: the PEST domain and the PDZ-BM [14]. This tail region is unstructured, and its flexible nature confers auto-inhibitory properties. The PEST domain constitutes the region from amino acid 352 to 399, which is rich in acidic (aspartate and glutamate) as well as serine and threonine residues. In contrast to the PEST domains in other signaling molecules, which normally promote protein degradation, the PEST domain of PTEN is associated with enhanced protein stability as its deletion has been shown to drastically decrease the PTEN protein expression. The C-tail region of PTEN is also populated by seven serine/threonine residues known to be phosphorylated by several key signaling molecules. These molecules include casein kinase II (CKII) that mediates phosphorylation at S370, S380, T382, T383, and S385; glycogen synthase kinase 3 beta (GSK-3β) that mediates phosphorylation at S362 and T366; and polo-like kinase 3 that mediates phosphorylation at T366 and S370 [30]. Among these phosphorylation sites, T366 appears to be an auto-dephosphorylation site, and its phosphorylation plays a role in tumor invasion [11].

### 2.5. PDZ-BM

The penultimate four amino acids of PTEN, namely ITKV, from residue 400 to 403 constitute the PDZ-BM, which is a short protein–protein interaction sequence that mediates the binding of PTEN to the PDZ domain containing signaling molecules frequently localized to the cell–cell junctions. PDZ domains are categorized into three classes. PTEN PDZ-BM has a shared consensus sequence, S/T-X-Φ-COOH, where X is any amino acid and Φ is any hydrophobic residues, with peptide ligands that bind to class I PDZ domains [31,32]. Physiological functions normally ascribed to PDZ domain proteins are mostly dynamic and transient in nature, such as synaptic transmission [33]. In total, 12 proteins are known to interact with PTEN through its PDZ-BM, namely hDLG, hMAST205, MAGI3, MAGI2, MAGI1, Bazooka/PAR-3, NHERF/EBP50, MPZ-1, PSD95, MAST2, PTPN13, and KIN-4. PTEN PDZ-BM is evolutionarily conserved, with similar sequences found in zebrafish, sea urchin, and *D. melanogaster*. NMR spectroscopy analysis of the binding between PTEN and MAST2-PDZ has revealed that although the last three amino acids of PTEN PDZ-BM, namely TKV, can account for 86% of the binding affinity, the phenylalanine residue at 392 can form hydrophobic interactions with residues in the β2, β3, and β5 strands of MAST2-PDZ [34]. Thus, these results indicate that PTEN PDZ-BM binds to the PDZ domain primarily through the C-terminal canonical motif, but also uses some distal N-terminal sequences.

## 3. Subcellular Distribution of PTEN

The Human Protein Atlas database (https://www.proteinatlas.org) reveals ubiquitous expression of PTEN in different organs (Figure 4). PTEN was previously believed to be localized in the cytoplasm. However, recent studies have shown that PTEN is also present in various subcellular compartments, such as the nucleus and mitochondria, and can even be secreted into the extracellular environment.

### 3.1. Cytoplasmic PTEN

Cytoplasmic PTEN converts PIP3 to PIP2, thus antagonizing PI3K/AKT pathway activation. Under normal conditions, only a small fraction of PTEN dynamically interacts with the plasma membrane [35]. PTEN can be activated and recruited from the cytoplasm to the inner face of the plasma membrane under some biological conditions to exert its anti-proliferative functions [15,35]. Cytoplasmic PTEN also plays an important role in facilitating apoptosis. Several mechanisms have been proposed. As a lipid phosphatase, cytoplasmic PTEN mainly suppresses the activation of the pro-survival kinase AKT, thus promoting the activation of a spectrum of pro-apoptotic genes such as GSK-3β, forkhead box O3a (FOXO3a), and caspase-9 [36,37]. Indeed, a positive correlation has been shown to exist between cytoplasmic PTEN and cell death in cancer cells [38]. As expected, the loss of cytoplasmic PTEN can lead to excessive PIP3 accumulation and the activation of a host of downstream signaling pathways, the overactivation of which can stimulate cell survival, growth, proliferation, angiogenesis, metabolism, and migration [39,40].

### 3.2. Nuclear PTEN

Accumulating evidence has suggested that compared with cytoplasmic PTEN, nuclear PTEN plays a totally different role in tumor suppression. Multiple clinical studies have detected nuclear PTEN in normal rather than cancer cells [41,42]. For example, the loss of nuclear PTEN has been documented in various cancers such as melanoma [43] and thyroid carcinomas [41]. An inverse correlation has been reported between nuclear PTEN expression and the mitotic index, suggesting that a lack of nuclear PTEN facilitates tumor cell proliferation [43]. Moreover, the expression level of nuclear PTEN has been used as a prognostic marker in various cancers [44,45]. At the cellular level, nuclear PTEN is critical for chromosome integrity, DNA repair, cell cycle arrest, and genomic stability. Accumulating evidence has suggested that nuclear PTEN functions as a guardian of chromosome integrity. Defective PTEN in mouse embryonic stem cells has been shown to cause genetic instability [46]. PTEN phosphatase activity is required for maintaining chromosome integrity [47] and preventing genomic alterations during cell division [48]. Nuclear PTEN can function as a mitotic phosphatase and physically interact with and dephosphorylate PLK1, thereby preventing polyploidy [48]. Cells lacking nuclear PTEN are hypersensitive to DNA damage, implying that PTEN plays an important role in DNA repair [49].

Multiple studies have revealed that nuclear PTEN can function as a brake for uncontrolled cell proliferation and regulate cell cycle progression. During the G1-S transition, nuclear PTEN downregulates cyclin D1 (CDK1) to inhibit G1 progression [50]. PTEN can also interact with p300 to maintain p53 acetylation, which, in turn, promotes PTEN–p53 interaction and regulates G1 arrest [51]. Moreover, nuclear PTEN activation has been shown to arrest G2/M progression. An overactivated Notch signaling pathway can lead to PTEN phosphorylation, thus promoting gastric tumorigenesis, whereas dephosphorylated nuclear PTEN can interact with a cyclin B1–CDK1 complex to arrest cells at the prometaphase [52]. Finally, DNA topoisomerase-2 alpha (TOP2A) mediates DNA decatenation and prevents chromatin entanglement and chromosome bridges during segregation. PTEN has been shown to physically interact with TOP2A to prevent its degradation [53].

### 3.3. PTEN in Cell Organelles

Emerging evidence has suggested that PTEN also plays a pivotal role in cell organelles other than the cytoplasm and nucleus. PTEN can function as a protein phosphatase in the endoplasmic reticulum (ER) to regulate ER-induced apoptosis. ER-localized PTEN physically competes with F-box/LRR-repeat protein 2 for type 3 IP3 receptor binding, which inactivates AKT and induces a subsequent ER-to-mitochondrial Ca^2+^ transfer, causing Ca^2+^-dependent apoptosis [54,55]. PTEN also mediates mitochondria-related apoptosis. One study showed a gradual accumulation of PTEN in the mitochondria of rat hippocampus during staurosporine-induced apoptosis. PTEN was found to increase cellular reactive oxygen species level and activate apoptotic cascades, whereas PTEN knockdown significantly rescued hippocampal cells from apoptotic damage [56]. Moreover, a recent study has suggested that PTEN plays a critical role in mitochondrial metabolism. This study reported differential mitochondrial oxidative phosphorylation states and bioenergetics in glioblastoma samples with different PTEN mutational statuses [57].

PTEN localized in the cell nucleolus also exerts tumor-suppressive activity. Nucleolar PTEN is essential for nucleolar homeostasis and morphology. PTEN knockdown has been observed to result in both quantitative and qualitative changes in nucleoli and increased ribosome biogenesis [58]. As increase in nucleolus and ribosome biogenesis is associated with increased cancer risk [59], nucleolar PTEN may exert its tumor-suppressive effect via the inhibition of ribosome biogenesis. Furthermore, a recent study has identified a PTEN isoform, PTENβ, which initiates translation from an AUU codon and has an extended 146-amino acid N-terminus. PTENβ has been found to be localized in the nucleolus where it regulates pre-rRNA synthesis by dephosphorylating nucleolin; however, its loss has been found to promote ribosome biogenesis [60].

PTEN has also been demonstrated to be enriched at the centrosomes and interact with the DLG1/EG5 motor protein complex during cell mitosis, thereby regulating proper mitotic spindle assembly and chromosome segregation [61]. PTEN may also phosphorylate Dishevelled, DVL, and participate in cilia disassembly and multicilia formation [62].

PTENα (also termed as PTEN-Long, PTEN-L) is a PTEN isoform whose translation initiates from a CUG codon upstream of the canonical start codon. This isoform has additional 173 (*Homo sapiens*) or 169 (*Mus musculus*) amino acids at the N-terminal region [63,64]. Immunofluorescence analysis of GFP-tagged PTENα has revealed that it is colocalized with the mitochondria, where it participates in mitochondrial energy metabolism by regulating cytochrome c oxidase activity. Subsequently, immuno-gold electron microscopy confirmed that PTENα is localized at the outer mitochondrial membrane. Functionally, PTENα impairs PRKN’s E3 ligase activity by preventing its mitochondrial translocation [64].

### 3.4. Secreted PTEN

Recent studies have reported that PTEN can be secreted from donor cells and taken up by recipient cells. This finding has revolutionized the concept that PTEN has only intracellular functions. PTEN can be packaged into exosomes and delivered under the control of NEDD4 family-interacting protein 1 (Ndfip1) with Ndfip1/Nedd4-mediated ubiquitination, thereby enhancing PTEN secretion. Exosomal PTEN is available for uptake by recipient cells, resulting in the repression of AKT activation and proliferation [65]. It has been demonstrated that PTENα can induce a complete tumor regression with a concomitant reduction of pAKT expression in a xenograft mouse model, implying PTENα can enter into neighboring tumor cells leading to tumor suppression [63]. PTENα is secreted extracellularly and exerts proinflammatory responses [66], thus suggesting its actions on immune cells. However, the detailed mechanism still needs further investigation. Exosomal PTEN has been considered as a therapeutic target for spinal cord injuries. Retinoic acid receptor β treatment induces the release of PTEN-enriched exosomes from neurons. Astrocytes that take up these exosomes will have reduced proliferation, leading to the inhibition of glial scar formation [67]. Moreover, PTENα can be secreted in the native form and be taken up by recipient cells. Secreted PTENα has been shown to be able to inhibit the PI3K pathway in a mouse model [63]. Several clinical studies have reported mutant forms of PTEN that could be detected in the biofluids of glioblastoma patients [68,69]. Reportedly, mutant PTEN proteins may act in a dominant-negative manner to suppress the function of wild-type PTEN through dimerization [70]. Thus, it will be of interest to determine whether cancer cells can secrete mutant PTEN to suppress wild-type PTEN function in recipient cells.

## 4. PTEN and Cancer Hallmarks

The link between *PTEN* and cancer was first established in 1997 when *PTEN* mutations were identified in multiple advanced tumors [3]. Overwhelming evidence has shown that *PTEN* loss of function occurs in a broad spectrum of human cancers. The highest percentage of *PTEN* aberrations has been found in uterine cancer, glioblastoma multiforme, and prostate cancer based on the data from the cBioPortal database (Figure 5A). This revealed the highest alteration frequency of *PTEN* in uterine cancer. Missense mutations account for the predominant genetic alteration in uterine cancer. However, deep deletions of *PTEN* are far more frequent in prostate cancer patients. Our speculation is that since *PTEN* loss in prostate cancer is associated with more advanced metastatic disease, the complete deletion of the *PTEN* gene may therefore be more prevalent [71]. On the contrary, in uterine cancer patients, *PTEN* is frequently mutated in Type I endometrioid carcinoma, which is associated with good prognosis [72]. Missense mutations, in this case, may have less deleterious effects on PTEN, and with its tumor suppressor functions being partially preserved.

*PTEN* loss of function is one of the most frequent events in cancers. The cBioPortal sequence data have underestimated the actual frequency of *PTEN* alterations. A meta-analysis has reported that the loss of PTEN protein was found in 78% glioblastoma patients and 48% endometrial tumor patients. Another study has revealed that deletions, including the *PTEN* locus in The Cancer Genome Atlas (TCGA database), have been identified in 143/170 (85%) of glioblastomas [73]. Even without the evidence of *PTEN* genetic changes, a considerable proportion of glioblastoma patients have reportedly shown reduced *PTEN* mRNA expression. Moreover, methylation of the *PTEN* promoter has been demonstrated to be a hallmark of cancers such as low-grade glioma and melanoma [74,75].

*PTEN* follows the “continuum model of tumor suppression” instead of the classical two-hit hypothesis, involving subtle expression changes that may influence tumor progression even without the loss of an allele [76]. Accumulating evidence has suggested that partial loss of PTEN function is sufficient for promoting tumor initiation and progression [77]. In addition, one study that used a hypomorphic *Pten* mouse model with reduced PTEN levels also demonstrated that subtle reduction in PTEN expression is enough to confer cancer susceptibility [78]. Dysregulation of PTEN expression can be attributed to multiple mechanisms, including transcription, miRNA or ncRNA targeting, and protein stability. For example, transcriptional silencing of *PTEN* by promoter hypermethylation has been reported in endometrial cancer, glioblastoma, and lung cancer [79,80,81]. Moreover, post-transcriptional changes in *PTEN* have also been revealed to be crucial in tumorigenesis [82].

It has been found that *PTEN* is concurrently mutated with specific genetic alterations such as *TMPRSS2-ERG* gene fusion and *TP53* mutation. *TMPRSS2-ERG* gene fusion and *PTEN* mutation have been revealed to drive prostate carcinogenesis cooperatively [83]. Indeed, the concurrence of *TMPRSS2-ERG* gene fusion and *PTEN* loss is associated with poor outcome [84]. The reason for this coexistence is unknown, but it has been pointed out that *TMPRSS2-ERG* fusion may facilitate the generation of *PTEN* deletions [85]. It has been reported that *TMPRSS2-ERG* alone is insufficient to drive tumorigenesis. Such collaboration may confer a selective advantage to promote precancerous lesions to aggressive cancer [86].

*TP53* is another gene mutated concurrently with *PTEN* in cancers of the prostate, cervix, and breast (Figure 5B). A combination of *Pten* and *Trp53* loss has been found to drive a prostate tumor progression in a mouse model [87]. Activation of PI3K pathway promotes MDM2-dependent p53 degradation, while ectopic PTEN overexpression can stabilize p53 by increasing its half-life [88]. In addition, PTEN can regulate *TP53* transcriptional activity [89], in turn, *PTEN* is a transcriptional target of p53 [90,91]. Overall, PTEN-p53 may form a self-reinforced circuit, the dysfunction of which may promote tumorigenesis.

### 4.1. PTEN and Oncogenic Signaling

The role of PTEN loss in tumorigenesis is highly complex, and hyperactivation of the PI3K pathway is clearly the major oncogenic signaling output. Indeed, PIK3CA, which encodes the α subunit of PI3K, is also frequently altered in various cancers [92,93,94]. The classical tumor suppressor function of PTEN is mainly dependent on its lipid phosphatase activity, which dephosphorylates PIP3 and thereby inhibits the phosphoinositide 3-kinase PI3K signaling pathway [15]. The activation state of the PI3K pathway is normally measured based on the levels of AKT phosphorylation, and aberrant AKT upregulation is frequently observed in both early and advanced cancers [95]. Activated AKT regulates downstream genes such as those of epidermal growth factor receptor, vascular endothelial growth factor (VEGF) receptors, mitogen-activated protein kinase (MAPK), caspase-9, and mammalian target of rapamycin (mTOR) [96,97]. Pathways related to all of these genes have been found to be essential for multiple biological processes, including cell survival, cell migration/invasion, and cell cycle progression. However, the consequences of AKT1, AKT2, and AKT3 ablation have been reported to be quite different. AKT1 knockdown has an anti-tumor effect, whereas AKT2 ablation can promote tumor growth and AKT3 ablation has little effect [98].

### 4.2. PTEN and Cell Cycle

The role of PTEN in cell cycle regulation has been widely studied. PTEN loss has been found to exert pro-tumorigenic effects through cell cycle dysregulation. In one study, *Pten* deletion in mouse astrocytes led to accelerated proliferation both in vitro and in vivo [99]. PTEN reintroduction to PTEN-null glioblastoma cell lines was found to suppress cell proliferation by inducing G1 arrest through p27Kip1 upregulation, which inhibited downstream cyclin-dependent kinase 2 activity [100]. In leukemia, PTEN expression reduced the proliferation of leukemic T cells through all phases of the cell cycle [101]. Furthermore, simultaneous inactivation of one *Pten* allele and one or more Cdkn1b (encoding p27Kip1) alleles accelerated neoplastic transformation and increased tumor incidence in a mouse prostate cancer model [102], implying that p27Kip1 plays a crucial role in mediating the tumor-suppressive effect of PTEN.

### 4.3. PTEN and Cancer Genome Stability

Nuclear PTEN is a guardian of genome integrity. Reportedly, PTEN loss is associated with aneuploidy in human primary breast cancer cells [46]. Nuclear PTEN localizes to the centromeres to maintain chromosome stability by physically interacting with CENP-C, an integral part of the kinetochore that is essential for proper chromosome segregation during mitosis [47,103]. Disruption of PTEN and centromeres can lead to chromosomal instability, which is a hallmark of cancer [47]. More recently, PTEN has been shown to regulate spindle pole architecture and movement by directly interacting with DLG1/EG5 through its PDZ-BM. PDZ-BM-lacking cells are prone to chromosome missegregation, and PDZ-BM-lacking mice are susceptible to lymphomas and breast cancer development [61,104].

### 4.4. PTEN and Cellular Energetics

Overwhelming evidence revealed that cancer cells undergo reprogramming of their metabolic pathways to sustain rapid proliferation and growth. Most cancer cells derive their energy from glycolysis instead of oxidative phosphorylation, an adaptive mechanism referred to as the Warburg effect [105]. A *Pten* transgenic mouse model has demonstrated PTEN to be a negative regulator of glycolysis [106]. Indeed, PTEN loss can lead to a plethora of metabolic changes, mostly through the activation of downstream PI3K-AKT pathway [107]. Haploinsufficiency of *Pten* has been demonstrated to hypersensitize insulin-stimulated glucose uptake both in vitro and in vivo [108]. The PI3K/PKB pathway is essential to maintain the normal glucose homeostasis, while the PTEN deficiency-induced PI3K-AKT activation is responsible for the translocation of Glucose transporter type 4 (GLUT4) [109,110]. PTEN loss induced PI3K-AKT activation also inhibits forkhead box protein O1 (FOXO1) and proliferator-activated receptor-γ (PPARγ), thus affecting hepatic gluconeogenesis [111,112]. Moreover, AKT activation promotes ATP hydrolysis, resulting in a compensatory increase in aerobic glycolysis through the upregulation of ENTPD5 [113]. PTEN loss also affects lipid and protein synthesis. Inactivation of PTEN promotes aberrant sterol regulatory element-binding proteins (SREBP)-dependent lipogenesis, thus driving metastatic progression in a mouse prostate tumor model [114]. mTORC1 activation due to PTEN loss upregulates 4E-binding protein 1 (4EBP1) and p70S6 kinase, which enhances pro-tumorigenic protein synthesis, contributing to tumor growth [115]. It has been reported that mTORC1 mediates S-adenosylmethionine decarboxylase 1 (AMD1) stability and affects polyamine synthesis, which is essential for the transformation of oncogenic metabolic program [116]. In addition, PTEN elevation in a super *Pten* mouse model can negatively regulate glutaminolysis and the Warburg effect, resulting in an anti-tumor metabolism in vivo [117].

### 4.5. PTEN and Metastasis

PTEN loss plays a vital role in tumor metastasis and invasion. Some clinical studies have revealed a high risk of tumor metastasis in patients with PTEN inactivation [118,119]. In addition, PTEN mutation has been identified as one of the most prevalent events in metastatic cancers [120]. The mechanism involved is highly complex, and PTEN alteration alone is insufficient to confer all metastatic traits [87,121]. In a mouse prostate cancer model, PTEN loss has been shown to function as a second hit for the activation of the oncogenic RAS/MAPK pathway [122]. Cooperation between PTEN inactivation and RAS activation has also been reported to drive melanoma metastasis [123]. In a *Pten*-null murine prostate cancer model, activated AKT could directly phosphorylate WHSC1 to prevent its degradation, and increased WHSC1 further enhanced AKT activity in a feedforward manner to promote prostate cancer metastasis [124]. In addition, several pathways involving NOTCH [125], BRAF [126], and SMAD4 [127] have been reported to cooperate with PTEN to trigger tumor metastasis. PTEN loss may also confer invasive capability. Some in vitro studies have revealed that PTEN impairs cell migration in both phosphatase-dependent and -independent manners [128,129,130]. Moreover, PTEN inhibition has been demonstrated to enhance tumor invasiveness in vivo [131].

### 4.6. PTEN and Angiogenesis

Angiogenesis is a biological process involving the generation of new blood vessels from preexisting vasculature, which is vital for normal tissue development and wound healing. Pathological angiogenesis is an important hallmark of cancer, which is a fundamental step in the transition from the benign state to the malignant state [132]. A previous study has reported that PTEN reconstitution can significantly suppress angiogenic activity via PI3K-dependent regulation in a nude mouse orthotopic brain tumor model [133], suggesting that PTEN also plays a role in controlling tumor-induced angiogenesis. In a zebrafish model, haploinsufficient PTEN has shown to result in enhanced VEGF expression and vessel hyperplasia [134]. Reintroduction of PTEN C2 domain also inhibited HepG2 induced-angiogenesis and VEGF expression both in vitro and in vivo, suggesting that PTEN can inhibit VEGF-mediated angiogenesis in a PI3K-independent manner [135]. PTEN loss in endothelial cell activates the Notch pathway, excessive activation of which can result in vascular hyperplasia [136].

## 5. Regulation of PTEN in Physiological and Pathological States

The multifaceted nature of PTEN tumor suppressor is regulated by the complex modulation of its transcription, translation, catalytic activity, subcellular distribution, and interactions with other signaling molecules. Because these topics have been reviewed in the past, we will only focus on more recent significant findings related to the mechanisms of PTEN regulation.

### 5.1. miRNAs and LncRNAs

MicroRNAs (miRNAs) and long-noncoding RNAs (LncRNAs) contribute to the regulation of PTEN protein level in many cancers at the post-transcriptional level. There are several miRNAs found to bind the 3′-UTR of *PTEN*, among them the most prominent one is miR-21. miR-21 is one of the most frequently overexpressed microRNAs in human cancer, which directly targets *PTEN* mRNA and negatively regulates PTEN protein level, thus promoting cell growth and metastasis [137]. Other oncomiR such as miR-23a [131], miR-26a [138], miR-92a [139], miR-130a [140], miR-205-5p [141], and miR-425 [142] are also reported to negatively regulate *PTEN* expression and activated PI3K-AKT signaling pathway, which positively contributed to tumor initiation, progression, and metastasis. Furthermore, the pseudogene of *PTEN*, *PTEN* pseudogene 1 (*PTENP1*), shares extensive sequence similarity with *PTEN* mRNA in regions that harbor miRNA target sites. Thus, it functions like a miRNA sponge, restoring the *PTEN* mRNA level and enhancing its tumor suppressor activity [143]. Overexpression or exosome transmitted *PTENP1* suppressed cancer cell proliferation and tumor progression [144,145,146].

### 5.2. Catalytic Activity

#### 5.2.1. PIP2

PTEN catalytic activity is regulated through an allosteric mechanism involving the binding of anionic phospholipids to the N-terminal PBM of PTEN [10,147]. This regulation is highly specific as the addition of only PI(4,5)P2 and PI(5)P can stimulate the catalytic activity of PTEN. Furthermore, only di-C8 fatty acid but not di-C4 fatty acid is active. Spectroscopic evidence suggests that PIP2 binding induces a conformational change associated with an increased α-helicity [148]. The residues within the PTEN PBM that mediate this allosteric stimulation are K13, R14, and R15 [10]. Notably, PTEN N-terminal PBM has been implicated in nuclear localization through the ubiquitination of K13 residue [22]. Whether ubiquitination at K13 affects PIP2 binding and/or enzymatic activation is not yet clear. It is possible that PIP2 binding and PBM ubiquitination are mutually exclusive. It is speculated that their relative contributions determine the extent of PTEN subcellular distribution on the cell membrane and nucleus.

#### 5.2.2. Phosphorylation

Phosphorylation of PTEN C-terminal tail region between residues 360 and 385 inhibits the phosphatase activity of PTEN. More than 10 intracellular kinases are known to mediate the direct phosphorylation of PTEN. For example, BCR-ABL interacts with CKII, and this complex suppresses the catalytic activity of PTEN through phosphorylation of its C-terminal tail region [149]. NMR has revealed that phosphorylation events in two clusters—namely S380–S385 (cluster I) and S361–S370 (cluster II)—are mediated by CKII and GSK-3β, respectively [150]. Phosphorylation in cluster I is an ordered event occurring in the following sequence: S385 > S380 > T383 > T382. For cluster II, the order is S370 > T366 > S362 > S361 > T363. Phosphorylated PTEN tail is believed to mask the N-terminal catalytic domain or C-terminal C2 domain through intramolecular interactions [16,151,152,153]. Screening for PTEN mutants displaying greater membrane-binding capacity has revealed that Q17, R41, E73, N262, and N329 residues mediate the intramolecular interactions with the C-tail [154,155]. Indeed, these residues are distributed in the catalytic pocket and membrane-facing surfaces of PTEN. Mutation of these residues significantly enhanced the intrinsic phosphatase activity of PTEN [154,155]. Overall, phosphorylation in the C-terminal tail region induced a closed, inactive conformation that prevented membrane binding, whereas reduction of the extent of phosphorylation resulted in an open, active confirmation. We profiled the mutational status of the 10 known phosphorylation sites between residues 336 and 401 in human cancers but detected only some sparing mutations (Figure 3). This was expected because mutations in these residues are predicted to enhance PTEN membrane binding and catalytic activities. Notably, we identified the putative phosphorylation sites at Y27, Y46, Y68, Y155, and Y174, which displayed greater mutational frequency. Their roles in PTEN function are not yet clear.

#### 5.2.3. Redox

Hydrogen peroxide-mediated oxidation inactivates PTEN catalytic activity by the formation of disulfide bridge between C124 and C71 in the catalytic pocket [156,157]. The reactivation of oxidized PTEN has been shown to be mediated by thioredoxin [156,158,159] and glutathione [160]. However, some derivatives of organic peroxides, such as the tumor-promoting *tert*-butyl hydroperoxide, have been shown to irreversibly oxidize and inactivate PTEN [161]. This irreversible redox regulation of PTEN may be a mechanism underlying the tumor-promoting actions of these agents. As per the COSMIC database, C124 and C71 have 29 and 13 missense mutations, respectively, in endometrial, breast, brain, prostate, and larger intestine cancers.

#### 5.2.4. Ubiquitination

A rather unusual observation is that the Ret finger protein (RFP), a E3 ligase of PTEN, mediates atypical ubiquitination of PTEN on multiple lysine residues in the C2 domain. Interestingly, these modifications do not alter PTEN protein turnover or localization but instead reduce its lipid phosphatase activity by more than 10-fold [162].

### 5.3. Membrane Targeting

The ability of PTEN to interact with the lipid bilayer is crucial for its tumor-suppressive activity. Using total reflection internal microscopy imaging techniques, it was estimated that PTEN interacts with the cell membrane for less than 200 ms [35]. This finding is consistent with the results of most immunofluorescence analysis studies showing that PTEN is preferentially localized to the cytoplasm instead of the cell membrane. However, this notion is disputed by the findings of super-resolution light microscopy, which showed that PTEN is in fact localized to endosomal vesicles tethered to microtubules through PI(3)P [163]. Again, the CBRIII motif in the C2 domain of PTEN mediates this interaction. Notably, PTEN displays sequence homology to auxilin, a protein essential for endocytosis, and is recruited to clathrin-coated vesicles [164].

The C-terminal tail region of PTEN plays an important regulatory role in controlling PTEN binding to the lipid bilayer. In general, the phosphorylated tail blocks PTEN from the membrane by physically interacting with the C2 domain via intramolecular interactions [153,165]. This assertion is supported by molecular dynamic simulation and has been validated by neutron reflectometry, which revealed the unstructured tail region tugging closely to the C2 domain, thereby blocking the membrane access region. In contrast, the tail region is repelled from the negative anionic phospholipid bilayer, thus allowing the C2 domain to bind to the membrane surface [165]. Random mutagenesis analysis in *Dictyostelium discoideum* revealed that mutant residues in the catalytic pocket (C124R), CBR3 (N262Y, K69E), Cα2 (N329H, N329I), and C-tail (Y379C/H/N, S380P/Y, D381V, T382I, T383I) enhanced the membrane-binding capacity, suggesting their involvement in an auto-inhibitory function [154]. Protein semi-synthesis and photo-crosslinking methods have demonstrated that each of the individual phosphorylation sites at S380, T382, T383, and S385 in the C-tail region contributed incrementally to the auto-inhibitory activity [166]. In the *D. discoideum* system, single-molecule imaging analysis revealed a ‘hopping’ mode of interaction between PTEN and the plasma membrane. This is mediated by the Cα2 helix of the C2 domain of PTEN [167]. As indicated above, PTEN-L is a long isoform of PTEN and is secreted into the extracellular space. Interestingly, hydrogen/deuterium exchange mass spectroscopy showed that PTEN-L has a membrane-binding element helix between residues 151 and 174 that alters the membrane-binding mechanism from the “hopping” to “scooting” mode [168]. These PTEN isoforms together with the complex combination of phosphorylation events generate diversity in the catalytic and membrane-binding capacities of PTEN proteins. Based on the COSMIC database, none of the above-mentioned residues except C124 are subjected to significant mutational alterations (Figure 3). This is expected because mutation would likely lead to greater membrane recruitment.

### 5.4. Stability

PTEN is a stable protein with reported half-live of >12 h [35,169]. The stability of PTEN is controlled by two key biochemical events: phosphorylation and ubiquitination. Numerous studies have demonstrated that the phosphorylation of PTEN C-tail confers stability presumably by rendering it in a closed conformation that can protect proteolysis-sensitive sites from protein degradation. Indeed, the substitutions of four frequently phosphorylated sites at residues S380, T382, T383, and S385 with alanine or the PTEN-4A mutant drastically reduced PTEN half-live by six fold [30]. Similarly, deleting the entire C-tail also reduced the stability of PTEN [35]. Indeed, mice carrying a *Pten* gene lacking the C-terminal tail region, *Pten^Δ^*^C^, harbored multiple tumors [170]. For ubiquitination, polyubiquitination at K13 and K289 has been intimately linked to its proteasome-mediated degradation [171]. Numerous E3 ligases and deubiquitinases have been identified and are being reviewed extensively by other groups [12,153,172]. A few E3 ligases for PTEN are of relevance to cancer. NEDD4 is the first known E3 ligase that mediates PTEN degradation [171,173]. NEDD4 overexpression has been observed in multiple human tumors [174]. Furthermore, antagonistic actions between NEDD4-1 and CK1α regulate PTEN stability in lung tumor growth [175]. CHIP, the chaperone-associated E3 ligase, binds to and mediates direct ubiquitination and degradation of PTEN. Indeed, the levels of PTEN and CHIP display an inverse relation in human prostate cancer [176]. Another NEDD4 family member, WWP2, also referred to as atropine-1-interacting protein 2, is another E3 ligase for PTEN [177]. The ability of WWP2 to degrade PTEN has been implicated in melanoma [178] and endometrial cancer development [179]. WWP2 is downregulated by Cdh1, another E3 ligase that drives M to G1 cell cycle progression [180]. In fact, WWP2-knockout mice have shown reduced body size and increased PTEN protein levels [181]. Finally, OTUD3, an ovarian tumor protease family member of E3 deubiquitinase, has been shown to increase PTEN stability [182]. Indeed, OTUD3 transgenic mice have shown reduced tumorigenic potential and high protein levels. In fact, human cancers harbor missense mutations in OTUD3 that abolish its ability to enhance PTEN protein levels. More recently, PTEN has been shown to drive a feedforward mechanism of upregulating the transcription of its own deubiquitinase USP11. USP11-deficient mice are susceptible to PTEN-dependent tumors [183]. Indeed, E3 ligase-targeting drugs are being developed to increase PTEN levels in human cancers [184,185,186]. Notably, numerous mutations in the PTEN gene associated with PTEN hamartoma tumor syndrome are known to affect protein stability without affecting the polyubiquitination sites [187].

### 5.5. Nuclear Targeting

Sequences responsible for the nuclear-cytoplasmic partitioning of PTEN are mainly confined to the N-terminal region (see Section 2.1. above) [188]. PPTMs by ubiquitination and sumoylation play critical roles in controlling the translocation, exclusion, and retention of PTEN in the nucleus. Monoubiquitination at K13 and K289 and sumoylation at K254 promote PTEN nuclear translocation and retention, respectively [22,49]. The E3 ligases responsible for PTEN ubiquitination are NEDD4-1 and XIAP [22,189], and PIASxα has been implicated as the E3 ligase responsible for PTEN sumoylation [190]. PTEN in the cell nucleus is deubiquitinated by the HAUSP enzyme and then excluded [191]. PTEN has also been shown to enter the nucleus through passive diffusion, which is mediated by the major vault protein [38,192,193]. Based on the COSMIC database, very few mutations have been noted at K254, K266, and K289 (Figure 3).

The nuclear-cytoplasmic partitioning of PTEN is modulated by a host of positive and negative regulators. The following factors promote PTEN nuclear translocation: Importin-11 [194], Ndfip1 [195], Grb2 [196], ATM [197], PERK [193], SDHD [198], Rab5/Ndfip1 [199], PNUTS [200], LKB-1 [201], oxidative stress [202], and Ran [188]. [149], acid ceramidase [203], free fatty acid-induced oxidative stress [204], genotoxic stress [49], ΔNp63α [205,206], NPM1 [207], and ATP [208]. In contrast, the following factors cause PTEN depletion in the nucleus: BCR-ABL Based on the reported roles of these factors in tumorigenesis, nuclear PTEN has been implicated to play an anti-tumorigenic role, but this role is very likely to be tumor type and stage specific. For example, in glioblastoma multiforme, *PTEN* mutations—such as K13E, L320S, and T277A—reduce nuclear accumulation of PTEN [209]. More suggestive in human chronic myeloid leukemia is BCR-ABL, which promotes PTEN exclusion [210]. As the functions of PTEN in the cell nucleus are phosphatase independent, its role in conferring genome stability may be important in its tumor-suppressive functions.

## 6. Conclusions

This review highlights the latest findings of multifaceted mechanisms in regulating PTEN functions. However, there are still considerable number of unresolved questions. For example, how the diverse subcellular functions of PTEN are being coordinated? Similarly, what is the relative contribution to individual cancer hallmarks from the loss of PTEN functions in the cytoplasm, nucleus, and other subcellular organelles during tumor progression? Also, is there crosstalk between different PTMs of PTEN and how they are regulated? From a translational standpoint, a significant fraction of human tumors still harbors a wild-type copy of *PTEN*. The possibility of enhancing its expression or catalytic activity will be an area of future research. In summary, a full understanding of the regulatory mechanisms of this key tumor suppressor may guide the future development of more effective therapeutics to restore PTEN anti-tumor activities.

## Figures and Tables

**Figure 1 cancers-11-01247-f001:**
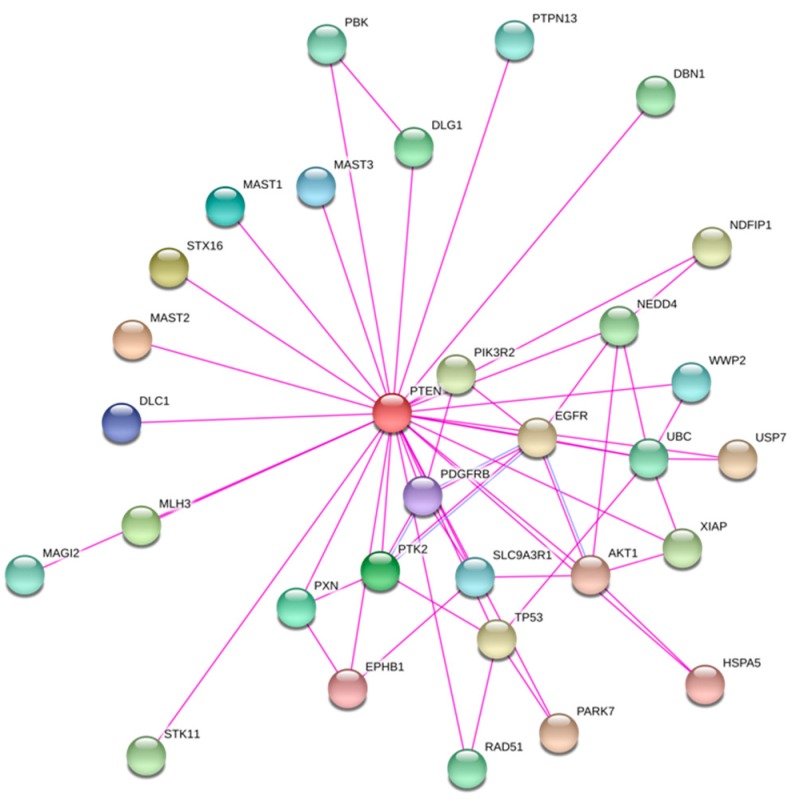
Interacting partners of PTEN. Protein interaction network of 30 representative interaction partners of PTEN (red dot) based on experimental evidence. Data downloaded from STRING functional protein association networks database (https://string-db.org).

**Figure 2 cancers-11-01247-f002:**
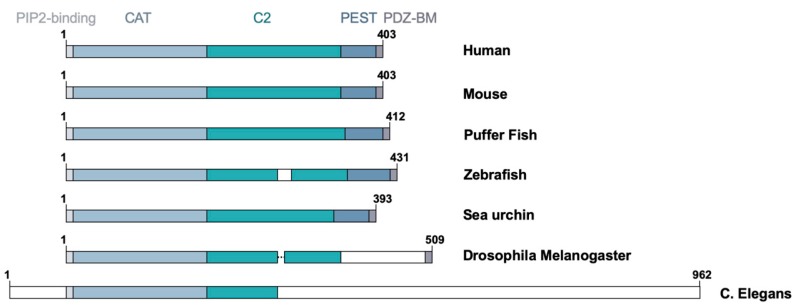
Functional domains in PTEN are evolutionarily conserved. Schematic representations of the five conserved domains in the species are indicated. Numbers indicate amino acid positions. Clear box, distinct sequence; dotted line, gap.

**Figure 3 cancers-11-01247-f003:**
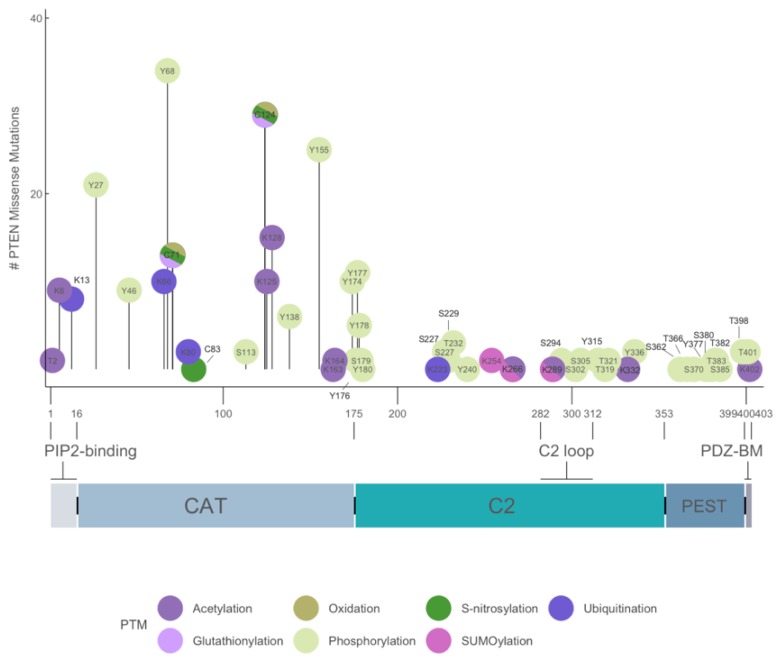
Mutation status of key PTEN post-translational modification (PTM) sites in human cancers. Schematic representation of mutation numbers at each site of the seven different types of PTMs known to PTEN protein. Individual PTMs are identified by indicated color codes (lower panel). Data derived from COSMIC database (https://cancer.sanger.ac.uk/cosmic). The domain structure of PTEN is shown. CAT, catalytic.

**Figure 4 cancers-11-01247-f004:**
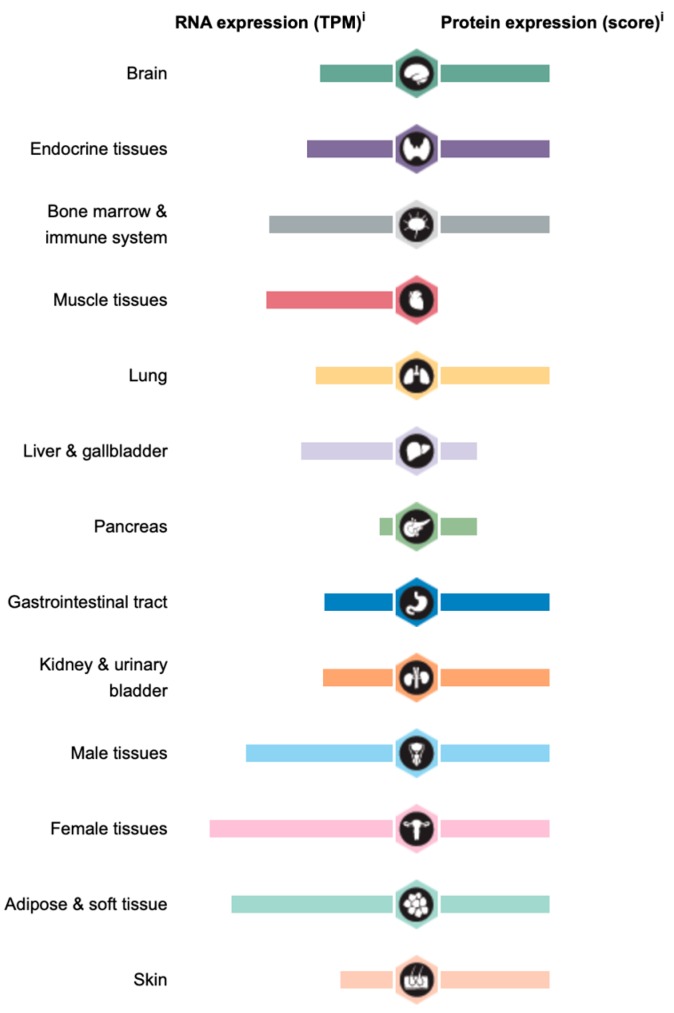
Tissue distribution of PTEN. Schematic representation of relative PTEN expression in indicated organs and tissues. Data obtained from The Human Protein Atlas database (https://www.proteinatlas.org).

**Figure 5 cancers-11-01247-f005:**
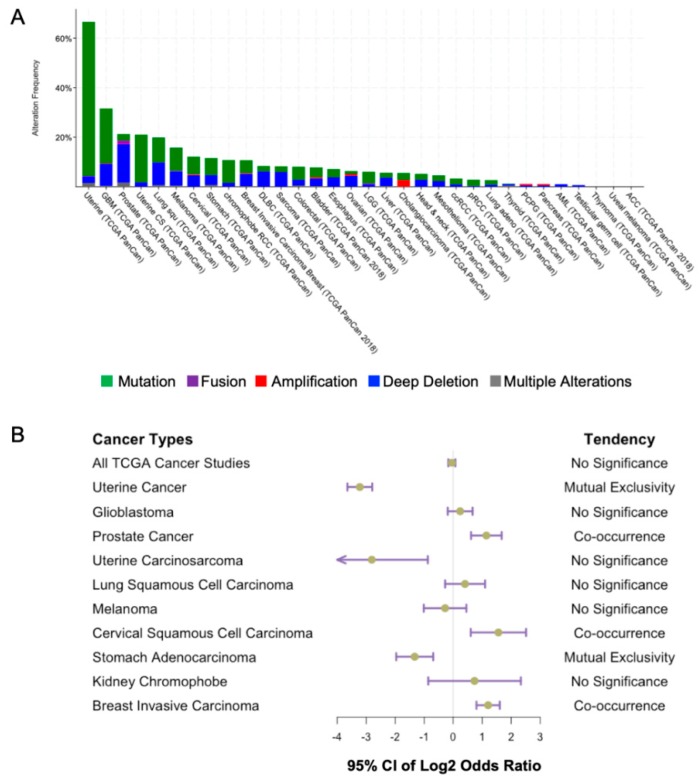
Alteration frequency of PTEN in human cancers. (**A**) Alteration frequency (%) of PTEN in indicated tumor types. (**B**) The extent of co-occurrence of *PTEN* and *TP53* mutations in different human tumors are shown. Data obtained from cBioPortal for Cancer Genomics database (https://www.cbioportal.org).

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
