# Peer review of "Multifaceted Regulation of PTEN Subcellular Distributions and Biological Functions"

_cancers, 2019, doi:10.3390/cancers11091247_

Round 1
Reviewer 1 Report
In the manuscript titled with “The multifaceted regulatory mechanisms of PTEN biochemical and biological functions”, the authors have tried to describe in diverse aspects the biochemical and biological functions of PTEN in association with cancer.
The authors should clarify the following problems in their revised manuscript.
First,
In the section 3.1 (with subtitle “PTEN and oncogenic signaling”), the authors should emphasize in deep detail about how PTEN/MMAC1 functions as a negative regulator of the phosphoinositide 3-kinase/Akt pathway (by using the reference 86).
Second,
The title of Figure 3 should be “Alteration frequency of PTEN in human cancers” because the authors have shown not only mutation frequency but also copy number (amplification and deep deletion) and fusion frequencies in the Figure 3.
Third,
Can the authors make insights onto why the copy number deep deletions of PTEN are more prevalent and causative in Prostate cancer patients than in Uterine cancer patients, whereas missense deleterious mutations are more predominant and causative in Uterine cancer patients (by using the Figure 3 data)?
Author Response
Reviewer 1
Critique 1: In the section 3.1 (with subtitle “PTEN and oncogenic signaling”), the authors should emphasize in deep detail about how PTEN/MMAC1 functions as a negative regulator of the phosphoinositide 3-kinase/Akt pathway (by using the reference 86).
Response: We have added a short paragraph in page 7 to detail the mechanism involved in suppressing the PI3K/AKT pathway by PTEN.
Critique 2: The title of Figure 3 should be “Alteration frequency of PTEN in human cancers” because the authors have shown not only mutation frequency but also copy number (amplification and deep deletion) and fusion frequencies in the Figure 3.
Response: We have made the suggested change to the title to the legend of Figure 5 (previously Figure 3).
Critique 3: Can the authors make insights onto why the copy number deep deletions of PTEN are more prevalent and causative in Prostate cancer patients than in Uterine cancer patients, whereas missense deleterious mutations are more predominant and causative in Uterine cancer patients (by using the Figure 3 data)?
Response: Our speculation is that since PTEN loss in prostate cancer is associated with more advanced metastatic disease, the complete deletion of the PTEN gene may therefore be more prevalent. On the contrary, in uterine cancer patients, PTEN is frequently mutated in Type I endometrioid carcinoma, which is associated with good prognosis. Missense mutations, in this case, may have less deleterious effects on PTEN, and with its tumor suppressor functions being partially preserved. We added this paragraph in page 6.
Reviewer 2 Report
Authors have provided an exhaustive review of the protein PTEN, its function and regulation, and role in cancer biology. There is a great discussion on the protein itself and its functional domains. Additionally, there is adequate discussion on PTEN enzymatic function, post-translational regulation, and subcellular localizations. These discussions of the PTEN protein are followed by a general discussion of the role of PTEN in cancer biology framed through the hallmarks of cancer. The last portion of the review is a more detailed discussion of the regulation of the protein that is very informative. The figures included are also informative and a helpful addition to the text. The following comments are intended to help the manuscript.
Optional: While the authors did point out PTEN follows a continuum model of tumor suppression, PTEN is frequently modified in conjunction with other gene modifications (e.g. TMPRSS2-ERG in prostate cancer). It may be worth mentioning some of these other “two-hit” members that are frequently altered with PTEN.Author Response
Reviewer 2
Critiques: Authors have provided an exhaustive review of the protein PTEN, its function and regulation, and role in cancer biology. There is a great discussion on the protein itself and its functional domains. Additionally, there is adequate discussion on PTEN enzymatic function, post-translational regulation, and subcellular localizations. These discussions of the PTEN protein are followed by a general discussion of the role of PTEN in cancer biology framed through the hallmarks of cancer. The last portion of the review is a more detailed discussion of the regulation of the protein that is very informative. The figures included are also informative and a helpful addition to the text. The following comments are intended to help the manuscript.
Optional: While the authors did point out PTEN follows a continuum model of tumor suppression, PTEN is frequently modified in conjunction with other gene modifications (e.g. TMPRSS2-ERG in prostate cancer). It may be worth mentioning some of these other “two-hit” members that are frequently altered with PTEN.
Response: We thank the reviewer for suggesting to include other “two-hit” members. We here include the classical two-hit tumor suppressor, TP53, in Page 7 and Fig. 5B.
Reviewer 3 Report
It is an important review aiming at understanding new mechanisms of PTEN in physiological and pathological conditions. It is well written, and the outlines of the paper are clear and logic. My comments are as the followings:
Major Points:
New approaches using microRNAs are currently being investigated towards mechanism of PTEN function. microRNAs play roles by repressing 3’ untranslated region (3’-UTR) of downstream target genes, such as PTEN. The 3’UTR of PTEN and the microRNAs regulating PTEN are not well described. In lines 264-265 of item 2.4 “Secreted PTEN”, the interaction between cancer cells and recipient cells via secreted PTEN needs further description. What types of cells are the recipient cells surrounding tumor cells? Are they immune cells and /or endothelial cells? What are possible effects of tumors cells impact on surrounding cells via secreted PTEN? In “PTEN and cancer hallmarks”, a section including PTEN and cellular metabolism should be added and discussed in cancer and other diseases.
Minor points
In the first sentence in the Abstract, “PTEN is a human tumor suppressor gene……”. Please delete “human” because PTEN is not only a human gene. In a subtitle of item 4. “Regulation of PTEN in normal and pathological states”, “normal” may be replaced with “physiological”.Author Response
Reviewer 3
Critiques: It is an important review aiming at understanding new mechanisms of PTEN in physiological and pathological conditions. It is well written, and the outlines of the paper are clear and logic. My comments are as the followings:
Major Points:
New approaches using microRNAs are currently being investigated towards mechanism of PTEN function. microRNAs play roles by repressing 3’ untranslated region (3’-UTR) of downstream target genes, such as PTEN. The 3’UTR of PTEN and the microRNAs regulating PTEN are not well described. In lines 264-265 of item 2.4 “Secreted PTEN”, the interaction between cancer cells and recipient cells via secreted PTEN needs further description. What types of cells are the recipient cells surrounding tumor cells? Are they immune cells and /or endothelial cells? What are possible effects of tumors cells impact on surrounding cells via secreted PTEN? In “PTEN and cancer hallmarks”, a section including PTEN and cellular metabolism should be added and discussed in cancer and other diseases.
Response: We have added an additional paragraph in section 4.1. miRNAs and LncRNAs as suggested by this reviewer.
For secreted PTEN, although not well defined, we added the potential targets cells and the impact of them in the tumor microenvironment in section 2.4.
For PTEN and metabolism, we have added a new paragraph as section 3.4. PTEN and metabolism as suggested by this reviewer.
Minor points
In the first sentence in the Abstract, “PTEN is a human tumor suppressor gene……”. Please delete “human” because PTEN is not only a human gene. In a subtitle of item 4. “Regulation of PTEN in normal and pathological states”, “normal” may be replaced with “physiological”.
Response: We have made all the edits as suggested by this reviewer.

Reviewer 4 Report
Tian Liu and colleagues in review titled “the multifaceted regulatory mechanisms of PTEN biochemical and biological functions” surveyed the current literature and presented a detailed report in PTEN biochemical and biological functions.
Minor: 1. Tittle need to be rephrased. The current title does not aligned well with text body.
Major:
Authors beautifully stated in the text the biochemical functions, structures and complexities associated with PTEN. The general readability will be greatly enhanced by including few more figures.
Need a detailed protein domain organization figure with functional domains (Lines 60-66) Need a regulation network figure where PTEN interacts with other cellular factors and how it functions (line 38-59). This may also include deregulation of PTEN in normal and pathological states PTEN orthologue figure showing conserved domain in different species (Line 67-70) Overall, this review is comprehensive, but it needs a future directions paragraph and a signature end statement with that a future hypothesis can be formed to decipher the complex network interactions of PTENAuthor Response
Reviewer4
Critiques: Tian Liu and colleagues in review titled “the multifaceted regulatory mechanisms of PTEN biochemical and biological functions” surveyed the current literature and presented a detailed report in PTEN biochemical and biological functions.
Minor: 1. Tittle need to be rephrased. The current title does not aligned well with text body.
We rephrased the title to “The multifaceted regulation of PTEN subcellular distributions and biological functions”
Major:
Authors beautifully stated in the text the biochemical functions, structures and complexities associated with PTEN. The general readability will be greatly enhanced by including few more figures.
Need a detailed protein domain organization figure with functional domains (Lines 60-66) Need a regulation network figure where PTEN interacts with other cellular factors and how it functions (line 38-59). This may also include deregulation of PTEN in normal and pathological states PTEN orthologue figure showing conserved domain in different species (Line 67-70) Overall, this review is comprehensive, but it needs a future directions paragraph and a signature end statement with that a future hypothesis can be formed to decipher the complex network interactions of PTEN
Response: The domain organization is added to both Figure 2 and 3. The interactome map of PTEN is now provided in Figure 1. PTEN orthologue is described in Figure 2. A future directions paragraph and a signature end statement are now incorporated in Page 11.

Round 2
Reviewer 4 Report
Thanks for incorporating changes. The review appears comprehensive.